# Photophysical Properties of Linked Zinc Phthalocyanine to Acryloyl Chloride:*N*-vinylpyrrolidone Copolymer

**DOI:** 10.3390/polym13244428

**Published:** 2021-12-17

**Authors:** Tamara Potlog, Ion Lungu, Pavel Tiuleanu, Stefan Robu

**Affiliations:** 1Physics Department and Engineering, Moldova State University, MD 2009 Chisinau, Moldova; ionlungu.usm@gmail.com; 2The Faculty of Chemistry and Chemical Technology, Moldova State University, MD 2009 Chisinau, Moldova; capitainvrungel@gmail.com (P.T.); stefan_robu@yahoo.com (S.R.)

**Keywords:** zinc phthalocyanine, *N*-vinylpyrrolidone, acryloyl chloride, ZnPc:*N*-VP:ClAC solution, absorbance, fluorescence emission, the lifetime of the excited state, dark cytotoxicity

## Abstract

This paper focuses on the linking of zinc phthalocyanine (ZnPc) to *N*-vinylpyrrolidone (*N*-VP): acryloyl chloride (ClAC) copolymer. The synthesis of binary *N*-VP:ClAC copolymer was performed by the radical polymerization method and then grafted to ZnPc by the Friedel Crafts acylation reaction. We have developed a water-soluble ZnPc:ClAC:*N*-VP photosensitizer with a narrow absorption band at 970 nm, fluorescence at λ_em_ = 825 nm and the decay fluorescence profile with 3-decay relatively longer times of 1.2 µs, 4.6 µs, and 37 µs. The concentration-dependent dark cytotoxicity investigated in normal fibroblasts (NHDF), malignant melanoma (MeWo), adenocarcinoma (HeLa), and hepatocellular carcinoma (HepG2) cell lines incubated to increased concentrations of ZnPc:ClAC:*N*-VP (up to 40 μM) for 24 h in the dark show low cytotoxicity. Maximum cell viability in HeLa and HepG2 tumor cell lines was observed.

## 1. Introduction

Currently, a variety of polymeric photosensitezers (PSs) have been developed by several research groups, especially for the conjugation of PSs with hydrophilic polymers which can greatly increase the solubility of PSs in aqueous solution [1,2,3,4]. The perspective of metallphthalocyanines (MPcs) in this area is due to several advantages over other organic materials, such as maximum absorption in the near-infrared region and high stability when exposed to light [5,6,7,8,9]. F. Setaro et al. prepared silicon phthalocyanine (SiPc) photosensitizers (PS) in polymeric micelles made of poly(ε-caprolactone)-*b*-methoxypoly (ethylene glycol) (PCL–PEG) block copolymers that efficiently photogenerate singlet oxygen and in vitro experiments demonstrate an excellent cellular uptake [10]. Feuser et al. [11] reported synthesis of ZnPc loaded poly (methyl methacrylate) (PMMA) nanoparticles (NPs) by mini-emulsion polymerization as PDT agents for leukemia treatment. Authors reported a series of polystyrene-*b*-poly(polyethylene glycol monomethyl ether acrylate) (PSt-*b*-PPEGA) [12] and poly(*N*-substituted acrylamide)-*b*-poly(polyethylene glycol monomethyl ether acrylate) (P(R)-*b*-PPEGA) [13] copolymers synthesized via RAFT as nanocarriers for ZnPc for PDT. H Shen et al. have been observed multiple changes in the aggregate morphologies of polystyrene-block-poly (4-vinylpyridine) (PS-b-P4VP) diblocks as a function of the apparent pH(pH*) in DMF/H_2_O mixtures [14]. Motloung et al. [15] synthetized and characterized tetra-pyridyloxy functionalized In (III), Zn (II) phthalocyanines and the tetra benzothiazole complexes into Pluronic F127 and Pluronic L121/F127 mixed micelles. Chiarante and his colleagues [16] found a similar effect with ZnPc derivative-loaded polymeric poloxamine micelles consisting of commercially available copolymer Tetronic^®^ 1107 (BASF, Ludwigshafen, Germany) for colon carcinoma treatment. This technique was also pioneered for coupling chlorin e6 to a dextran polymer and then conjugating the PS-loaded dextran to the tumor cells for over production of reactive oxygen species (ROS) to kill them [17]. The incorporation of the photosensitizers and suitable electron donors or electron acceptors into a polymeric chain is useful for various molecular system based on photo-induced electron transfer [18,19,20,21,22,23]. As we know, *N*-vinylpyrrolidone and acryloyl chloride are coupling elements, and can be obtained by radical polymerization in solution. The higher molecular weight type products are polymerized in aqueous solution using acryloyl chloride as initiator. Copolymers especially with monomers such as *N*-vinylpyrrolidone and acryloyl chloride compounds may also be produced by solution polymerization.

However, as far as we know there is no report in the literature on poly (*N*-vinylpyrrolidone)—acryloyl chloride copolymer grafted onto ZnPc to be used in photodynamic therapy. The research efforts in the present study were focused on binding zinc phthalocyanine to water soluble *N*-VP:ClAC copolymer and the investigation of the photophysical properties. Furthermore, the dark cytotoxicity are described.

## 2. Preparation and Characterization of Photosensitizer

### 2.1. Materials

*N*-vinylpyrrolidone (*N*-VP) with average mol wt. 111.14, acryloyl chloride (ClAC) (≥97% purity, containing ~400 ppm phenothiazine as stabilizer), Zinc (II) phthalocyanine (ZnPc), (97% purity) were purchased from “Aldrich”, Sigma-Aldrich, Inc., St. Louis, MO, USA. Azo-bis-isobutyronitrile (AIBN, Aldrich, Burlington, MA, USA) was recrystallized three times from ethanol. The other reagents were of analytical grade from commercial origin (Alfa Aesar, Heysham, UK) and used without further purification.

#### 2.1.1. *N*-VP:ClAC Synthesis

*N*-VP:ClAC copolymer was obtained by radical copolymerization initiated by azobisisobutyronitrile (AIBN) according to Figure 1. For example, 0.82 mL (0.010 mol) of ClAC and *N*-VP in quantity of 8.3 mL (0.089 mol, ClAC—10 mol. %) were dissolved in chloroform in a test tube in presence of 2 mol.% of AIBN. The glass tube was sealed and maintained under inert atmosphere at 80 °C for 2 h. The product is re-precipitated in methanol, filtered, and dried at 40 °C. Products with different contents were synthetized.

#### 2.1.2. ZnPc: *N*-VP:ClAC Synthesis

The synthetized copolymer of the composition n-mol.% *N*-VP and m-mol.% ClAC was conjugated with ZnPc molecule through a substitution reaction in the benzene nucleus using the Friedel Crafts acylation reaction in the presence of aluminum trichloride as the catalyst. For this 0.2 g of AlCl_3_ (1.4 mmol) were dissolved in a three-necked flask in 10 mL of chloroform at temperature of 0–5 °C with vigorous stirring. Further, while stirring, a mixture of copolymer *N-VP:ClAC* 0.15 g (1.4 mmol) in 5 mL of chloroform were introduced by a drip funnel during 15 min, followed by 0.2 g (0.35 mmol) introduction of ZnPc in chloroform by the same funnel. The reaction is maintained for 3 h. The obtained product is washed with methanol and then it is dried at 40 °C. The crude product was purified by column chromatography with silica gel eluting with chloroform. Products with different contents of ZnPc were obtained by varying its concentration. The FTIR transmittance spectra were recorded on an Alpha Bruker spectrophotometer in transmission mode (Bruker Corporation, Nasdaq: BRKR, Billerica, MA, USA) with a spectral range of 4000 to 400 cm^−1^. The appearance in FTIR spectra of ZnPc:*N*-VP: ClAC of new bands at ν = 1580–1605 cm^−1^ indicate the presence of *N*-VP nuclei as well as of aromatic nuclei of ZnPc [24].

#### 2.1.3. Physical Characterization Methods

The UV-Vis spectra of the solutions were measured using a UV-Vis spectrophotometer (Lambda 25, Perkin Elmer Inc., Shelton, CT, USA) from 200 nm to 1200 nm in 10 mm quartz cuvettes. The steady-state fluorescence spectroscopy was performed using a spectrometer (LS55 Perkin Elmer, Inc., Shelton, CT, USA) equipped with double grating excitation and emission monochromators. Time-Correlated Single Photon Counting (TCSPC) was used to determine the fluorescence lifetime. The time-resolved fluorescence spectra were recorded on a spectrometer (Edinburgh FLS980, Livingston EH54 7DQ, Oxford, UK). All emission decay profiles were determined in a 10 × 10 mm^2^ quartz cell, excited with a xenon lamp as the light source. For the analysis of fluorescence lifetime decays was used the multiexponential model [25]
(1)$$I\left(t\right)=\overset{M}{\underset{i=1}{\sum }}A_{i}exp\left(-\frac{t}{\tau_{i}}\right),$$
where I(t) represents the emission intensity at time *t*, A**_i_ and $\tau_{i}$ are the amplitudes and decay times of the *M* exponential components of the fluorescence decay. The mean decay time is given by $\langle \tau_{e}\rangle =\sum A_{i}\tau_{i}$. The best-fitting parameter values are found for reduced chi-squared values close to one, and the weighted residuals uniformly distributed around the zero line. ^1^H NMR (400 MHz) spectra were collected on Bruker Advance III 400 MHz spectrometer (Bruker Corporation, Nasdaq: BRKR, Billerica, MA, USA) with the DMSO *deuterated* solvent. All the measurements were made at room temperature (295 ± 1 K).

### 2.2. Preparation of Cell Culture

#### 2.2.1. Cell Culture

Cell were incubated in alpha-MEM medium (Lonza, Basel, Switzerland) supplemented with 10% fetal bovine serum (FBS, Gibco, Thermo Fisher Scientific, Waltham, MA, USA) and 1% penicillin-streptomycin-amphotericin B mixture (10 K/10 K/25 μg, Lonza, Basel, Switzerland).

#### 2.2.2. Dark Cytotoxicity

Cytotoxicity evaluations were performed on the normal dermal fibroblasts (NHDF, PromoCell, Heidelberg, Germany), malignant melanoma (MeWo), adenocarcinoma (HeLa), and hepatocellular carcinoma (HepG2) (CLS Cell Lines Service GmbH, Eppelheim, Germany) cells. The MTS assay was used to assess cytotoxicity. The advantage of MTS over other assay is that it is more soluble and nontoxic, allowing the cells to be returned to culture for further evaluation. Dark cytotoxicity was measured using the CellTiter 96^®^ AQueous One Solution Cell Proliferation Assay (Promega, Madison, WI, USA), according to the manufacturer instructions and ISO 10993-5 [26]. For each experiment, control cultures, treated with DMSO only, were established and processed in parallel. The final DMSO concentration was 1.25% and 0.625% for 40 µM and 20 µM samples, respectively. The stock solutions (considered to have 1.6 mM concentrations) were sonicated and vortexed before dilution in complete culture medium. The solution samples were dissolved in the culture medium. The cells were incubated for 24 h in the dark. After the medium in each well was replaced with 50 µL culture medium + 10 µL MTS reagent, then the absorbance was read at 490 nm on a FLUOstar^®^ Omega microplate reader (from BMG LABTECH, Ortenberg, Germany). The experiments were performed in triplicate and treated cells’ viability was expressed as a percentage of control cells’ viability. The results of the MTS assay were expressed as mean ± SD.

## 3. Results and Discussion

### 3.1. Absorbance and Fluorescence Spectra

Absorption and emission spectra of ZnPc-*N*-VP:ClAC were investigated in DMSO/H_2_O solution in 1:1 ratio. Figure 1a,b shows the absorption spectra of ZnPc-*N*-VP:ClAC in DMSO/H_2_O solution with the ZnPc various concentrations. The ZnPc-*N*-VP:ClAC concentration was selected during preliminary experiments so that the solution optical density would be less than 0.5, i.e., obeyed the Lambert–Bouguer–Beer law:$$\;D_{\lambda }=\varepsilon_{\lambda }\times C\times l,$$
where $D_{\lambda }$ is the optical density at wavelength λ of a solution of the substance, $\varepsilon_{\lambda }$ is the molar absorption coefficient of the substance at λ wavelength, *C* is the molar concentration of the substance, and l is the thickness of the solution absorbing layer (cm).

For a better understanding of ZnPc:*N*-VP:ClAC electronic absorption spectra in the Figure 1b is presented the absorbance of the *N*-VP:ClAC copolymer, ZnPc and ZnPc:*N*-VP:ClAC photosensitizer in DMSO/H_2_O. For all measurements of the absorbance as the baseline the DMSO/H_2_O solution in 1:1 ratio were taken. The *N*-VP:ClAC copolymer absorbs strongly in the UV wavelength region. The Q bands of the peripherally substituted ZnPc:*N*-VP:ClAC system are red-shifted when compared to the unsubstituted ZnPc in the same DMSO/H_2_O solvent. The spectra of ZnPc:*N*-VP:ClAC in DMSO/H_2_O solution show that the Q bands are 295 nm highly shifted to the red region of the spectra when grafted to *N*-VP:ClAC copolymer compared to ZnPc. We suppose that introducing the peripheral substituent *N*-VP:ClAC onto the macrocycle of ZnPc led to a significant bathochromic shift of the absorption spectra due to an increased destabilization of the HOMO electron state versus the LUMO state. The spectra of ZnPc:*N*-VP:ClAC in DMSO/H_2_O provided a broad electronic absorption Q band situated between 570 nm and 900 nm and the narrow band with maximum at 960 nm. We suppose that Q absorption band could be assigned to the π–π* transition on the Zn phthalocyanine macrocycle and the narrow band could correspond to the molecular charge transfer between ZnPc and *N*-VP:ClAC copolymer with electron donor–acceptor interaction [27,28].

The fluorescence emission spectra of ZnPc:*N*-VP:ClAC in DMSO/H_2_O are shown in Figure 2.

The fluorescence spectra of ZnPc:*N*-VP:ClAC excited at wavelengths of 750 nm showed broader and red-shifted strongest fluorescent properties at λ_em_ = 825 nm for the compound with 20 mol.% of ZnPc. The broadness and unstructured fluorescence of ZnPc:*N*-VP: ClAC are consistent with the typical features of charge-transfer emission.

The fluorescence decay time was investigated and the fluorescence lifetimes were calculated. Fluorescence lifetime refers to the average time a molecule stays in its excited state before it returns to its ground state by emitting. The emission decay at λ_ex_ = 750 nm represents an exponential fitting of the experimental data associated with the three lifetimes τ_1,_ τ_2_, and τ_3_, as shown in Figure 3. The time evolution of excited state signals for the ZnPc:*N*-VP:ClAC solution revealed the relatively longer fluorescence lifetimes 1.2 µs, 4.6 µs, and 37 µs, respectively. We suppose that we deal with the triplet–triplet annihilation upconversion process that involves multistep downward energy transfer processes, in which the most energies are lost in intersystem crossing and triplet–triplet energy transfer. Anti-Stokes fluorescence occur via radiative transitions to make the ions come back to the ground states [29]. The observed fluorescence emission of ZnPc:*N*-VP:ClAC solution originates from the charge-transfer transition of donor–acceptor segments attached to the peripheral substituent *N*-VP:ClAC onto the macrocycle of ZnPc. The results of the time-resolved fluorescence spectra studies clearly justify testing them in photodynamic processes.

So, we have developed a water-soluble ZnPc: *N*-VP:ClAC photosensitizer with high absorption at 970 nm, the excitation/emission wavelength of 740/825 nm and with the relatively longer fluorescence lifetimes of 1.2 µs, 4.6 µs, and 37 µs compared to fluorescence lifetime of other copolymer linked to phthalocyaninephotosensitizers in solution at room temperature [30,31].

### 3.2. Cytotoxicity Analysis

The results obtained in our evaluation of the cytotoxicity of different ZnPc:*N*-VP:ClAC solution concentrations against NHDF, MeWo, HeLa, and HepG2 cells can be seen in Figure 4. In vitro cytotoxic study of 40/20/10/5/2.5/1.25 µM of ZnPc:*N*-VP:ClAC concentrations on normal fibroblasts (NHDF), malignant melanoma (MeWo), adenocarcinoma (HeLa), and hepatocellular carcinoma (HepG2) cell lines indicates low cytotoxicity towards the all mentioned tumor cells. Maximum cell viability after 24 h of incubation at all above mentioned ZnPc:*N*-VP:ClAC concentrations in HeLa and HepG2 tumor cell lines was observed. However, decreased cell viability is seen in MeWo tumor cell at higher concentration, while when the ZnPc:*N*-VP:ClAC concentration decreased an increase in cell viability is revealed. Generally, the activity of ZnPc:*N*-VP:ClAC photosensitizer was not significant dependent on the concentration.

### 3.3. Structural Determination

The ^1^H-NMR spectra give the information about the types of protons, number of each type of proton and their environments. For the production of copolymers having anionic functional groups, monomers, such as acryloyl chloride, were used. The^1^H-NMR spectrum from Figure 5 (red curve) shows that the formation of *N*-VP:ClAC copolymer is due to hydrogen bonding.

The ^1^H-NMR spectrum of the ZnPc:*N*-VP:ClAC complex presented in Figure 5, blue line, indicates that the signals δ = 1–3 ppm are attributed to C–CH_2_–C and NCH_α_ fragments of pyrrolidone in the copolymer chain, while 7.0–9.5 ppm peaks are characteristic for aromatic system of phthalocyanine substituted rings (12H ZnPc: *N*-VP-ClAC).

To visualize the surface structural morphology and identify the associated material elemental compositions of the synthesized materials, scanning electron microscopy (SEM) in conjunction with energy dispersive X-ray spectroscopy (EDX) was carried out. The EDX analysis presented in Figure 6 confirms the presence of ZnPc.

The presence of Al in the quantity indicated in the inset table from Figure 6 does not affect the dark cytotoxicity of the photosensitizer. Probably, the synthesized product was not sufficiently purified.

## 4. Conclusions

We synthesized a water-miscible form of ZnPc grafted to copolymer of *N*-vinylpyrrolidone and acryloyl chloride system. Based on the absorbance at 970 nm, remarkable fluorescent properties at λ_em_ = 825 nm and the relatively longer fluorescence lifetimes of the excited states 1.2 µs, 4.6 µs, and 37 µs we believe that ZnPc:*N*-VP:ClAC system may be a promising novel photosensitizer for innovative PDT. Following PDT, about 80% cell viability was observed for malignant melanoma MeWo at investigated concentration of 40 μM. The low dark toxicity of the ZnPc:*N*-VP:ClAC in tumor cells is promising and further studies on the synthesis of a water-miscible ZnPc:*N*-VP:ClAC system is essential, in order to improve the photophysical, photochemical properties and the dark toxicity.

## Data Availability

Not applicable.

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
