# Peer review of "Photophysical Properties of Linked Zinc Phthalocyanine to Acryloyl Chloride:N-vinylpyrrolidone Copolymer"

_polymers, 2021, doi:10.3390/polym13244428_

Round 1
Reviewer 1 Report
The manuscript is poorly presented. Besides broken English, it is difficult to understand the sense of statements. The manuscript is sloppy, unprofessional, has a bunch of nonsense. The main confusion is between lambda (ex) and wavelengths of emission. I present both principal and minor critical comments below.
Nonsense starts with the title, lines 2-4: ….”properties”…..:…”process”
Line 11: “Friedel-Kravts method”. Probably they meant Friedel-Crafts acrylation reaction
Lines 13-14: Here and in the manuscript text below: impossible to understand at what excitation wavelength a emission is observed. Do they deal with an anti-Stokes shift? What means relatively long time of emission (fluorescence?). Compared to what? To a common lifetime of fluorescence of organic molecules in solution at room temperature?
Line 16: “MTS techniques” is not defined.
Line 14: “relaxation time”. Do they mean here and in the manuscript below a life-time of an excited state?
Line 104: “MPcs” are not defined
Line 18: should be “absorption”. “Fluorescence emission”? Fluorescence is an emission.
References: some names, some words are underlined. Why?
Line 11: “N-vinylpyrrolidone (average mol wt 138 40,000)”. Its mol weight is 111 g/mol in fact.
Line 200, Fig. 1b. Is the excitation wavelength 750 nm? It is” hidden” somewhere below in the text. What is the ration of DMSO:H2O?
Line 226, Figure 2. Should be named something like “Decay kinetics…” Why namely two life-times are selected? Why not one or not four?
Line 266: “Based on the absorbance at 970 nm, remarkable fluorescent properties at λem = 825 nm and longer lifetime of the excited state…”(?) That is the most important point but it is written an incoherent way. Did the authors observe an anti-Stokes shift? Why does fluorescence have a lifetime of microseconds?
PS. I recommend the authors to read:
Cite this:DOI: 10.1039/c6cs00415f
Anti-Stokes shift luminescent materials for bio-applications
Xingjun Zhu, Qianqian Su, Wei Feng* and Fuyou Li*
Author Response
Dear reviewer,
Please find attached here response to your questions and comments.
Thank you for your time.
Authors

Reviewer 2 Report
Dear Editor, dear Authors, Tamara Potlog et al. submitted a paper on the synthesis of water soluble copolymer based on N-vinylpyrrolidone (N-VP) and acryloyl chloride (ClAC) via radical polymerization. The obtained N-VP:ClAC copolymer was then complexed to a hydrophobic zinc phthalocyanine (ZnPc) photosensitizer by the Friedel-Crafts method. The resulting zinc phthalocyanine copolymer ZnPc:ClAC:N-VP was found to be water soluble and a good photosensitizer that exhibit a narrow absorption band at 970 nm, and a fluorescence emission maximum at 825 nm following an excitation at 750 nm, and relatively longer-lived excited states relaxation times. Furthermore, the authors have tested the ZnPc:ClAC:N-VP toxicity effect at different concentrations on dermal fibroblasts (NHDF), malignant melanoma cells (MeWo), adenocarcinoma (HeLa), and hepatocellular carcinoma (HepG2) using MTS assay. They have demonstrated that the ZnPc:ClAC:N-VP copolymer has no significant cytotoxicity towards the tumor cell lines. The manuscript is of interest. However, I believe that the manuscript not well written and results might be discussed in a much better way. Therefore, the manuscript needs a major revision prior to acceptance for publication in Polymer MDPI Journal. I have the below comments to the authors
Comments to the authors:
- Abstract “…and then grafted to ZnPc by the Friedel-Kravts method.” Please correct here it should Friedel-Crafts not Kravts, no !.
- Abstract last two words, use MTS assay instead of MTS technique.
- The authors should summarize in the abstract their results from cell viability tests.
- Page 1, Introduction line 104, “Currently, MPcs are widely used for PDT.” The authors should write the complete name/definition for MPcs and PDT followed by the abbreviations.
- Page 1, line 116 “ The authors [31]” give the name of the first author and et al. of the reference 31 instead of the authors. Similar for other citation in the manuscript for example page 1, line 119 “Other authors [32,33]’’ and in the rest of the manuscript.
- Page 1, line 118 “superior 1O2” use a complete definition singlet oxygen (1O2)
- Page 1 line 123 “ A shift of 8–10 nm for the Q-bands”you should clarify further here by defining and indicating the position in nm of the initial Q bands.”
- Page 1, lines 123, 124 “ …the highest quantum yield of ΦΔ.” Delete of
- Page 1, lines 126, 127 why you use underline/link for photosensitizer and polymeric ?
- Page 2, lines 137, 138 revise the font size/representation for average mol wt 40, 000. Also here why there is a link to Sigma Aldrich.
- Page 2, line 142 “azo-bis-isobutyronitrile (AiBN)” use AIBN instead of AiBN for abbreviation.
- Page 2, line 145 “Friedel-Kravts method” Please correct here it should Friedel-Crafts method.
- Page 3, Figure 1 a, why the authors spectra show a highest absorbance for the 20 mol % ZnPc (violet spectrum), while at higher concentration 30 mol% the spectrum is found below the 20 mol% and in betweem 10 and 15 mol% (orange spectrum). While the authors claim the applicability of the Beer’s law, please explain.
- Page 3, lines 204, 205 “The spectra of ZnPc:N-VP:ClAC in DMSO/H2O solution shows that the Q bands are 204 highly shifted to the red region of the spectra when grafted to N-VP:ClAC copolymer.” The authors should mention here, which spectra are talking about, where is the Q band as well as by how much the shift is exactly and what was the initial position of the Q band before grafting to to N-VP:ClAC copolymer. Please revise and clarify further.
- Page 3, line 215, why the word unstructured is underlined and has a web link ? Please revise.
- Page 5, figure 4, “Furthermore, 1H NMR spectrum of ZnPc:N-VP:ClAC revealed a new singlet signal 253 at δ 4.66 ppm due to -OCH2 as well as triplet and quartet signals at δ 2.04, 2.09, 2.35 and 254 2.43 ppm representing OCH2-CH3, respectively.” What the author mention here is not very clear in the figure and not supported for example the singlet at 4.66 ppm is not appearing in the figure, also the triplet and quartet signals for me look more as multiplets. Please revise to clarify further.
- Why EDX is showing a strong peak for Aluminium, please explain.
Sincerely Yours,
Author Response
Dear reviewer,
Please find attached here response to your comments and questions.
Thank you for your time.
Authors

Round 2
Reviewer 1 Report
In my opinion, it is possible to publish the revised manuscript.